# Generating information-dense promoter sequences with optimal string packing

**Virgile Andreani** [1,2], **Eric J. South** [2,3], **Mary J. Dunlop** [1,2,3]*

**1** Biomedical Engineering Department, Boston University, Boston, Massachusetts, United States of America,
**2** Biological Design Center, Boston University, Boston, Massachusetts, United States of America,
**3** Molecular Biology, Cell Biology & Biochemistry Program, Boston University, Boston, Massachusetts, United States of America

☯ These authors contributed equally to this work.
* mjdunlop@bu.edu

**Data Availability Statement:** The solver is implemented in an open-source library available at: https://gitlab.com/dunloplab/dense-arrays. The code used to generate the data and make the figures from this paper is available on the "paper"

## Abstract

Dense arrangements of binding sites within nucleotide sequences can collectively influence downstream transcription rates or initiate biomolecular interactions. For example, natural promoter regions can harbor many overlapping transcription factor binding sites that influence the rate of transcription initiation. Despite the prevalence of overlapping binding sites in nature, rapid design of nucleotide sequences with many overlapping sites remains a challenge. Here, we show that this is an NP-hard problem, coined here as the nucleotide String Packing Problem (SPP). We then introduce a computational technique that efficiently assembles sets of DNA-protein binding sites into dense, contiguous stretches of double-stranded DNA. For the efficient design of nucleotide sequences spanning hundreds of base pairs, we reduce the SPP to an Orienteering Problem with integer distances, and then leverage modern integer linear programming solvers. Our method optimally packs sets of 20–100 binding sites into dense nucleotide arrays of 50–300 base pairs in 0.05–10 seconds. Unlike approximation algorithms or meta-heuristics, our approach finds provably optimal solutions. We demonstrate how our method can generate large sets of diverse sequences suitable for library generation, where the frequency of binding site usage across the returned sequences can be controlled by modulating the objective function. As an example, we then show how adding additional constraints, like the inclusion of sequence elements with fixed positions, allows for the design of bacterial promoters. The nucleotide string packing approach we present can accelerate the design of sequences with complex DNA-protein interactions. When used in combination with synthesis and high-throughput screening, this design strategy could help interrogate how complex binding site arrangements impact either gene expression or biomolecular mechanisms in varied cellular contexts.

## Author summary

The way protein binding sites are arranged on DNA can influence the regulation and transcription of downstream genes. Areas with a high concentration of binding sites can enable complex interplay between transcription factors, a feature that is exploited by

branch of this repository. All dense array sequences generated in the manuscript are available in the repository. The /benchmarks folder provides detailed information for each dense array.

**Funding:** This work was supported by the National Science Foundation (MCB-2143289 and MCB-2324909 to MJD). EJS received support from the National Institutes of Health under award number T32GM130546. The funders had no role in study design, data collection and analysis, decision to publish, or preparation of the manuscript.

**Competing interests:** The authors have declared that no competing interests exist.

natural promoters. However, designing synthetic promoters that contain dense arrangements of binding sites is a challenge. The task involves overlapping many binding sites, each typically about 10 nucleotides long, within a constrained sequence area, which becomes increasingly difficult as sequence length decreases and binding site variety increases. We introduce an approach to design nucleotide sequences with optimally packed protein binding sites, which we call the nucleotide String Packing Problem (SPP). We show that the SPP can be solved efficiently using integer linear programming to identify the densest arrangements of binding sites for a specified sequence length. We show how adding additional constraints, like the inclusion of sequence elements with fixed positions, allows for the design of bacterial promoters. The presented approach enables the rapid design and study of nucleotide sequences with complex, dense binding site architectures.

## Introduction

The layout of binding sites within nucleotide sequences can define both biomolecular interactions and subsequent biological processes. For example, the architecture of cis-regulatory regions—stretches of non-coding DNA containing binding sites for transcription factors and other regulatory proteins—plays a key role in controlling the expression of downstream genes. These regions serve to recruit regulators and convert cellular signals into transcriptional outputs [1–4]. Regions with few binding sites are thought to enable transcription factors to bind and then permit higher overall transcription levels, even when these proteins are in limited quantities [5]. Conversely, dense clusters of binding sites can elicit complex phenomena such as cooperative binding effects [6–9], steric hindrance [10,11], and transcription factor sharing [5]. These emergent properties can lead to nonlinear transcriptional regulatory logic, wide dynamic ranges, and context-sensitive gene expression [12–14].

As the complexity of cis-regulatory regions becomes more evident, researchers have begun to employ forward engineering approaches to systematically map promoter sequences to transcriptional readouts. One of the most prominent methods in this regard is the use of massively parallel reporter assays (MPRAs) [15–17]. These assays link the expression of a reporter gene to a specific cis-regulatory variant, which is often situated upstream on either an episomal or genomically integrated locus [18]. Traditionally, the design of promoter variant libraries for MPRAs has followed one of two different strategies: either diffuse nucleotide diversification, achieved through methods like error-prone PCR or random mutagenesis [19,20], or hybrid engineering approaches that fuse core promoter subregions with other discrete binding site elements [21,22]. In hybrid engineering approaches, the design of sequence variants for MPRAs has typically focused on adjusting binding site spacing and consensus sequences, often placed in adjacent or proximal positions, while overlooking the potential for overlapping binding sites [15,23–26]. While these studies offer insights into how binding site positioning and biophysical constraints inform promoter strength, their general omission of overlapping binding sites has limited the characterization of nonlinear, emergent properties that can arise in natural densely arranged cis-regulatory regions.

Designing nucleotide sequences with overlapping binding sites becomes challenging when the total length of the desired binding sites, each typically about 10 nucleotides long [27], surpasses the fixed length of the intended output sequence. The overall problem increases in complexity as the pool of binding sites expands, causing an exponential increase in potential binding site configurations. Studies that have tackled the design of overlapping sequences, and

their effects on transcriptional regulatory logic, often rely on *ad hoc* methods or generate a limited set of short sequences [26,28–30], thus restricting the generation of large libraries. Meanwhile, generative AI techniques are starting to show promise in emulating the complexity of context-dependent promoters [31–37]. Many of these models are trained on large datasets of natural sequences [38–40], leading to synthetic promoters that mimic these natural examples. However, these models may struggle to generate sequences with cis-regulatory logic that is not present in natural genomes, as they tend to produce sequences within their training distribution. The choice of training data significantly shapes the model, mirroring assumptions about the sequence distributions to explore [41]. Consequently, there is growing interest in using synthetic DNA to generate training data to facilitate the discovery of novel expression responses [42].

Here we present a novel computational method for the design of nucleotide sequences with densely packed DNA-protein binding sites, which we name the nucleotide String Packing Problem (SPP), related to the classical Shortest Common Superstring problem in theoretical computer science [43]. We deliberately use the terminology "string" instead of "sequence" in the problem name to avoid confusion with the concept from computer science of a "subsequence," which is not necessarily contiguous. This distinction is crucial for distinguishing the SPP, where an individual binding site must not be split, from a potential "Sequence Packing Problem" where the sequence AGGA of length 4 would fit the three elements AA, GG, AGA (among others), due to the non-contiguous nature of subsequences.

After proving that the SPP is NP-hard, we reduce it to the Orienteering Problem with integer distances, which is an optimization problem related to the Traveling Salesman Problem [44]. This reduction allows us to formulate nucleotide string packing as an integer linear programming problem, which can then be solved efficiently with a variety of open-source and commercial solvers. This formulation not only underscores the computational complexity of the design task but also lays the groundwork for generating sequences that accommodate a maximal number of protein binding sites. The expressivity of integer linear programming also allows for a variety of design objectives and constraints, which can be used to tailor the specifics of the output nucleotide sequence. We demonstrate how a modification to the model makes it possible to adjust the order of solutions returned to favor a more uniform representation of binding sites across sequences. We also demonstrate that by adding additional constraints to the SPP, like the inclusion of sequence elements with fixed positions, we can effectively interweave dense regions of binding sites around -35 and -10 sigma factor recognition sites. This approach enables the design of complex bacterial promoters that interact with multiple sigma factors. The integer linear programming model and all its extensions have been made available to the community as part of an open-source Python library available at https://gitlab.com/dunloplab/dense-arrays. Equally applicable to large-scale libraries and more focused studies, this computational approach serves as a resource for designing nucleotide sequences with complex DNA-protein binding architectures.

## Results

### Complexity of the String Packing Problem (SPP)

The problem of packing DNA-protein binding sites into a DNA sequence of a fixed length can be formulated more generally as an optimization problem on strings: consider a finite alphabet $\Sigma$, a collection $R$ of strings from $\Sigma^*$, and a natural integer $L$. The goal is to find a string $w$ of length $L$ that maximizes the number of different strings of $R$ contained in $w$, that is, the number of strings $x \in R$ such that there exists $w_0 \in \Sigma^*$ and $w_1 \in \Sigma^*$ such that $w = w_0 \times w_1$ (Fig 1A).

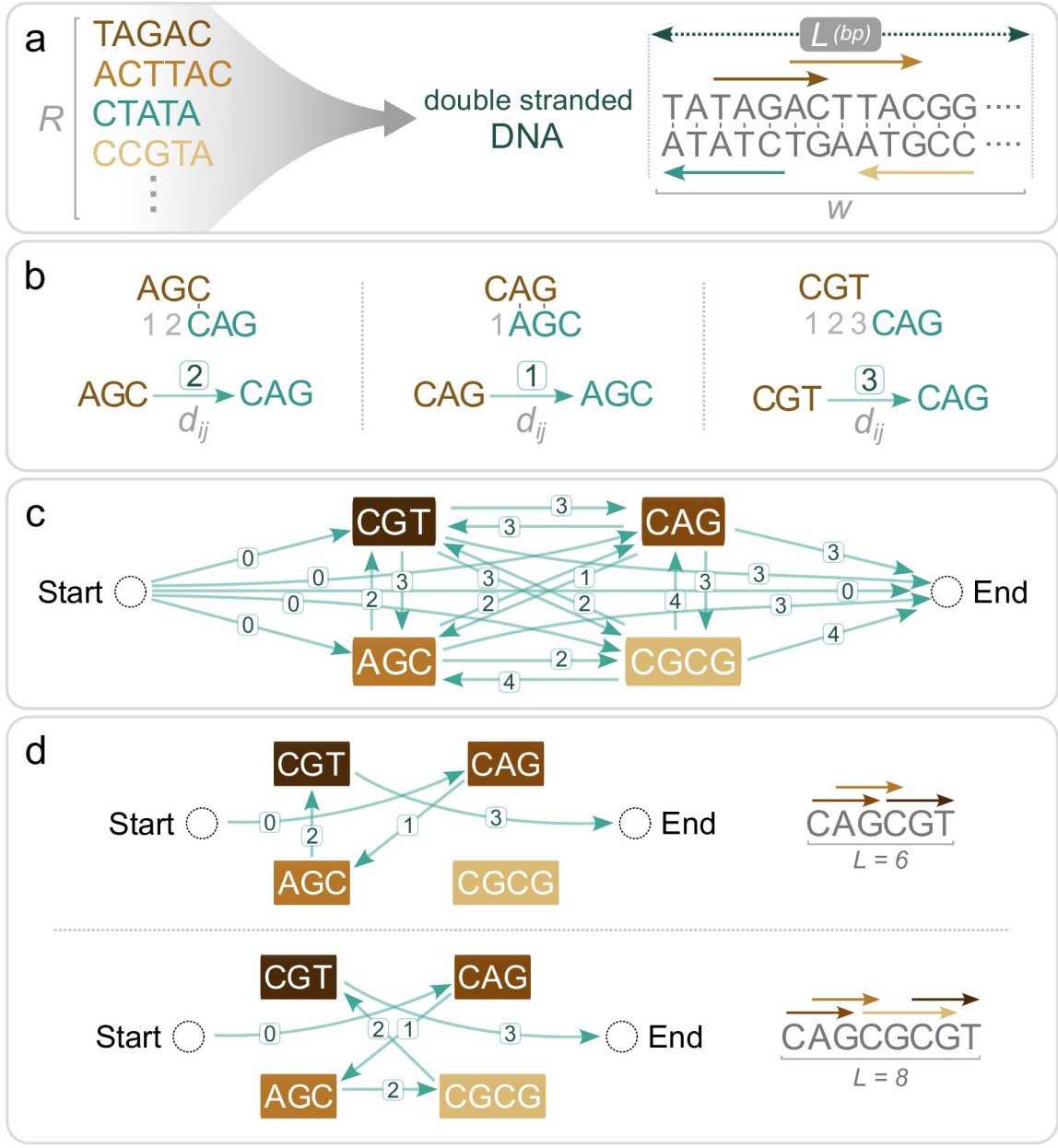

**Fig 1. Formulation of the nucleotide String Packing Problem (SPP) as an Orienteering Problem (OP).** (A) The SPP consists of fitting as many different strings from a binding site collection $R$ as possible inside a string of a given length $L$. bp, base pairs. (B) We define an asymmetrical metric between strings $i$ and $j$, $d_{ij}$, as the least number of shifts needed for the prefix of the second string to match the suffix of the first string. (C) After adding Start and End vertices, we construct the complete graph between binding site strings with the associated shift, $d_{ij}$, between strings shown. The OP is solved on this graph. For the double-stranded problem, we also include the reverse-complements of all the binding sites, omitted here for clarity. (D) Representative solutions for two different values of $L$.

To determine the complexity of this problem, let us first consider the associated decision problem (SPP-DECISION): given the alphabet $\Sigma$, the collection of strings $R$, two natural integers $L$ and $N$, is there a string of length $L$ or shorter that contains $N$ or more of the strings of $R$? We will show that this problem is NP-complete, that is, it belongs to the set of NP problems at least as hard as all other NP problems.

To show that SPP-DECISION is NP-complete, we first need to show that it is an NP problem, and then that it is at least as hard as all other NP problems. (1) To show that it is an NP problem, we need to show that we can construct certificates of positive answers, with sizes that are polynomial in the size of the input, which can then be verified by a polynomial-time algorithm. (2) To show that SPP-DECISION is at least as hard as all other NP problems, we will proceed by reduction from another NP-complete problem: the decision problem corresponding to the Shortest Common Superstring problem (SCS-DECISION) [45] (as described in [43]). The proof follows:

1. SPP-DECISION is in NP: A certificate of existence of a superstring can be a list of the strings included in it with their offsets from the beginning of the superstring. The size of this certificate is polynomial in the size of the problem input. Moreover, this certificate can be verified by a polynomial-time algorithm. SPP-DECISION is then in NP.

2. SCS-DECISION is reducible to SPP-DECISION: The SCS-DECISION problem is closely related to the SPP-DECISION problem, in that given $\Sigma$, $R$, and $L$, the question is to determine if there exists a string of length $L$ or shorter that contains all strings from $R$. Considering any SCS-DECISION problem described by $(\Sigma, R, L)$, we can build an SPP-DECISION problem described by the same alphabet $\Sigma$, collection of strings $R$, and integer $L$, and where $N = |R|$. A positive (respectively negative) answer to this SPP-DECISION problem implies a positive (respectively negative) answer to the original SCS-DECISION problem, which shows that SCS-DECISION is reducible to SPP-DECISION, and thus that SPP-DECISION is NP-complete.

Let us now consider the original optimization problem, SPP, which is not a decision problem. We will show that it is NP-hard, that is, every problem in NP is reducible to it. To show this, we will show that SPP-DECISION, which we just showed is NP-complete, is reducible to the SPP. Let us consider any SPP-DECISION problem, described by $(\Sigma, R, L, N)$. We can construct the SPP described by $(\Sigma, R, L)$. Let us call $M$ the number of different substrings (i.e., binding sites) of $R$ included in the optimal solution to this problem. It is enough to compare $M$ to $N$ to answer the SPP-DECISION problem, thus showing that SPP-DECISION is reducible to SPP and therefore that SPP is NP-hard.

## Performance and scalability

Brute force approaches to solving the SPP are not scalable, due to the NP-hardness of the task. No polynomial-time algorithms are known to solve it optimally, and the only deterministic algorithms known all eventually rely on an exponential number of steps in the worst case. But the choice of the approach greatly impacts the practical solve time. For example, for a binding site collection with $|R| = 50$ binding sites and a sequence of length $L = 200$ base pairs, one brute force approach could be to enumerate all possible $4^{200} \approx 10^{120}$ sequences, which is computationally infeasible. Despite their computational challenges, many NP-hard problems hold significant practical value. This has led to the development of specialized solvers designed to efficiently manage these problems, although they may face difficulties with extremely large instances [46]. Many numerical solvers have been developed to tackle the NP-hard integer linear programming problem, which restricts linear programming to integer variables. We formulated the SPP as an integer linear programming problem by reducing it to a variant of the Travelling Salesman Problem (TSP) known as the Orienteering Problem (OP) [44,47,48] (see Methods for the integer linear model). This formulation abstracts away the notion of nucleotides and considers instead the interaction between binding sites through a shifting metric defined between every pair of binding sites (Fig 1B), leading to a dramatic decrease in the

complexity of the problem. Indeed, a brute force approach to solve this formulation would now only need to enumerate all possible paths in the graph visiting each binding site at most once (Fig 1C and 1D). For the binding site collection considered above ($|R| = 50$ and $L = 200$), there are about $10^{65}$ such paths (as calculated using the approach in Ref. [49]), which is still astronomically large, but much less than $10^{120}$. Thus, the same problem can be formulated in different ways, and the amount of computational work needed to solve them can be very different. The algorithms in integer linear programming are also exponential but are designed to dramatically decrease the exponential constant. While they cannot cheat the exponential for very large tasks, they embed heuristics that help them to quickly find an optimal solution in many cases. Here, thanks to their internal algorithms and heuristics, the integer linear programming solvers do not have to explore all $10^{65}$ paths of the OP, and thus are able to find a solution for a problem of size $|R| = 50$ and $L = 200$ highly efficiently, returning solutions within seconds. The strength of our approach relies on two factors: the formulation of the initial SPP as a graph problem abstracts the notion of nucleotides and eliminates the need to represent them individually, which drastically cuts down the search space, and the use of an optimized solver further reduces the complexity by tackling this problem using state of the art algorithms.

Solve times for the SPP are expected to be dependent on the parameters of the problem (size of the binding site collection, $|R|$, and length of the sequence, $L$). To investigate the influence of these parameters, we considered collection sizes, $|R|$, between 10–100 distinct binding sites, each with a random length between 5 and 15 base pairs, where these lengths are based on known protein-DNA interactions [27]. We also considered typical lengths of cis-regulatory regions between 20–300 base pairs [50] to use as our sequence length, $L$. Using the highly efficient Gurobi solver as a backend for the integer linear programming, we found that the solve times were rapid (Fig 2A). For example, the collection with $|R| = 50$ binding sites and a sequence length of $L = 200$ base pairs can be solved in 2 seconds on a laptop. We found that solve time scales proportionally with the number of variables of the model, that is, with the square of the binding site collection size. The solve time is also largely independent of the length of the sequence (Fig 2B), which may be explained by the fact that this parameter does not change the number of variables or the structure of the model. It is also possible that increasing the length of the sequence may increase the number of optimal solutions, making it easier for the solver to find one of them, despite their increasing complexity. The solve times range from fractions of seconds for collections of size $|R| = 20$, to 10 seconds for collections of size 100, making this is an efficient and accessible method for practical nucleotide design tasks.

## Heuristic approaches to solve the problem

In some cases, it is not necessary to reach the optimal solution to an NP-hard problem if a suboptimal but sufficiently good solution is acceptable. It can be much easier to find suboptimal solutions, and this can often be achieved by "approximation algorithms," many of which run in polynomial time. For instance, in the case of the TSP, the nearest neighbor algorithm—always visiting the closest city not yet visited—operates in time proportional to the square of the number of cities, yet it may yield solutions significantly worse than the optimal. For metric TSPs (i.e., when the distance respects the triangle inequality), the Christofides–Serdyukov algorithm is a polynomial-time algorithm that is guaranteed to return a solution no longer than 1.5 times the optimal solution [51–53]. Approximation algorithms also exist for the SCS problem [54]. The greedy algorithm (repetitively merging the two strings with the best overlap, until there is only one remaining) will return a superstring at most 3.425 times longer than the optimal string. A recent approximation algorithm allows this factor to be lowered to 2.475

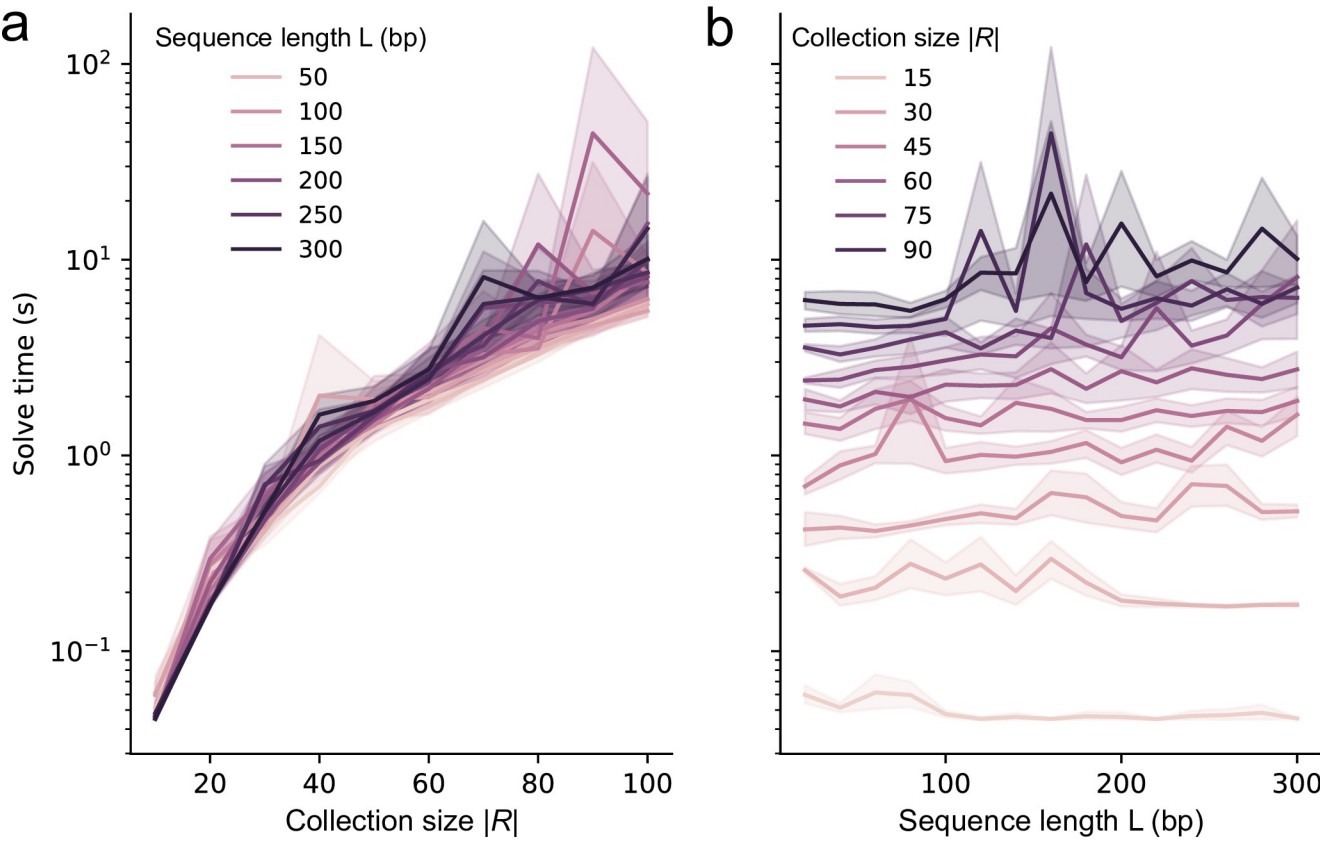

**Fig 2. Performance of the SPP.** Solve time as a function of (A) binding site collection size, $|R|$, and (B) sequence length, $L$. All experiments were run 10 times, with the Gurobi v10.0.1 backend. The shaded regions represent the bootstrapped 95% confidence interval around the mean. All replicates use a different, randomly sampled binding site collection, with random binding sites of uniformly random length between 5 and 15 base pairs. The binding site collection size and sequence length are specified on the figure. Double-strand optimization is performed here. For all these experiments, Gurobi was given access to an Intel Xeon E5-2650 v2 CPU (16 logical threads) and its memory usage was under 3 GB. A timeout of 600 seconds was set, reached only 8 times out of 1500, which we excluded from these data.

[55]. Thus, if adapted to our problem, a similar strategy would create an array in the worst case less than half as dense as it could be.

To test if an approximation method could produce results comparable to the optimal results returned by our integer linear programming approach, we employed a greedy algorithm. This algorithm involves repetitively merging the two binding sites with the best overlap until only one is left (in the worst case this is 3.425 times longer than the optimal solution). If this resulting sequence is shorter than the targeted sequence length, while fitting all binding sites in the collection, it is an optimal solution and is thus returned. If the sequence extends beyond the target length, we apply a sliding window corresponding to our target length, and then locate the sequence segment containing the highest number of binding sites; this segment is subsequently returned. In a numerical study using randomly generated sequences to represent binding sites, we compared the number of binding sites in the solutions returned by this greedy algorithm to the optimal number computed with our integer linear programming formulation. We found that the approximation algorithm often returned fewer binding sites than optimal (60% of the optimal in the worst cases), with large binding site collection sizes ($|R|$) and compact sequence lengths ($L$) posing challenges that only the integer linear programming formulation was able to tackle (S1 Fig). Meta-heuristics—general strategies applicable to a range of problems—are an alternative approach and have shown promise in identifying quality

suboptimal solutions. Genetic algorithms, for instance, have shown efficacy in similar nucleotide overlapping problems [28,29]. However, the practical application of these algorithms is limited by difficulties in tuning them to yield good solutions quickly.

## Diversity of the generated solutions

Next, we returned to our integer linear programming approach and quantified the diversity of sequence solutions returned by the algorithm. Some applications might require the generation of multiple sequences, composed of binding sites from the same collection, for the purposes of library curation or biological screening. For example, generating diverse DNA sequences from the same binding site collection can enable nuanced studies in functional genomics, or facilitate comparative analysis during either drug screens or synthetic biology circuit optimizations.

We found that maximizing the number of sites in a sequence introduces a bias for smaller binding sites (S2 Fig). Interestingly, we found that this bias is not exclusively tied to size variation, as it manifests even among equally sized binding sites. To demonstrate this, we created random binding site collections of $|R|$ = 10 binding sites of 10 base pairs each. We enumerated the best solutions for a final sequence of length $L$ = 50 base pairs: in all cases, six binding sites were fit in this sequence at best (score = 6) (Fig 3A). For the three random collections we tested, the number of different sequences realizing this best score were 1854, 2654, and 10,922, where the higher numbers correspond to a case where there were more ways to create sequences composed of 6 binding sites, for example due to similar sequences in the starting binding site collection. If the frequency of binding site inclusion was perfectly uniform, we would expect each binding site to occur with a frequency of 0.1 (1/$|R|$). However, we found that every binding site was not equally likely to be part of these best solutions (Fig 3B). To rule out the chance that these findings were simply due to sampling variations, we used a control experiment where we randomly selected six binding sites from the collection as many times as there were solutions, removing the constraint that these sites needed to fit into a sequence. To compare the sampling bias between these two scenarios, we sorted binding sites by decreasing frequency (Fig 3B and 3C). The flatter the line, the more uniform the distribution is. Conversely, a steeper slope indicates a less diverse distribution, tending to favor certain binding sites over others. We found that the creation of dense arrays inherently tends to favor specific binding sites, shaped by the unique interplay among sites within each specific binding site collection (Fig 3D). We next investigated the reasons for discrepancies in the representation of binding sites that were all the same length, focusing on results across the top scoring dense arrays. We compared each binding sites' average shift metric $d_{ij}$ to and from other binding sites, representing their average overlap with other binding sites, with their rate of appearance in top-scoring solutions. As expected, we found that the binding sites with the highest average overlap with others were more represented in the top-scoring solutions (S3 Fig).

Bias in binding site selection due to size (S2 Fig) or sequence (Fig 3D) can potentially be transiently amplified by solver-specific tendencies: when faced with solutions of equal score and no additional constraints, the solver is free to generate outcomes in an order that reflects its intrinsic algorithms, often more akin to local exploration than direct sampling. Indeed, different solvers return the same solutions of equal score in different orders (S4 Fig). This implies that the distribution of the binding sites involved in the first few solutions of equal score (for instance, those containing 6 binding sites), may not accurately represent the distribution of the entire set. To create a sample of the best solutions with a more balanced distribution of binding sites, one could ask the solver for a much larger number of solutions and filter this set to homogenize the representation of binding sites. However, this is computationally wasteful, as many of the generated solutions are discarded.

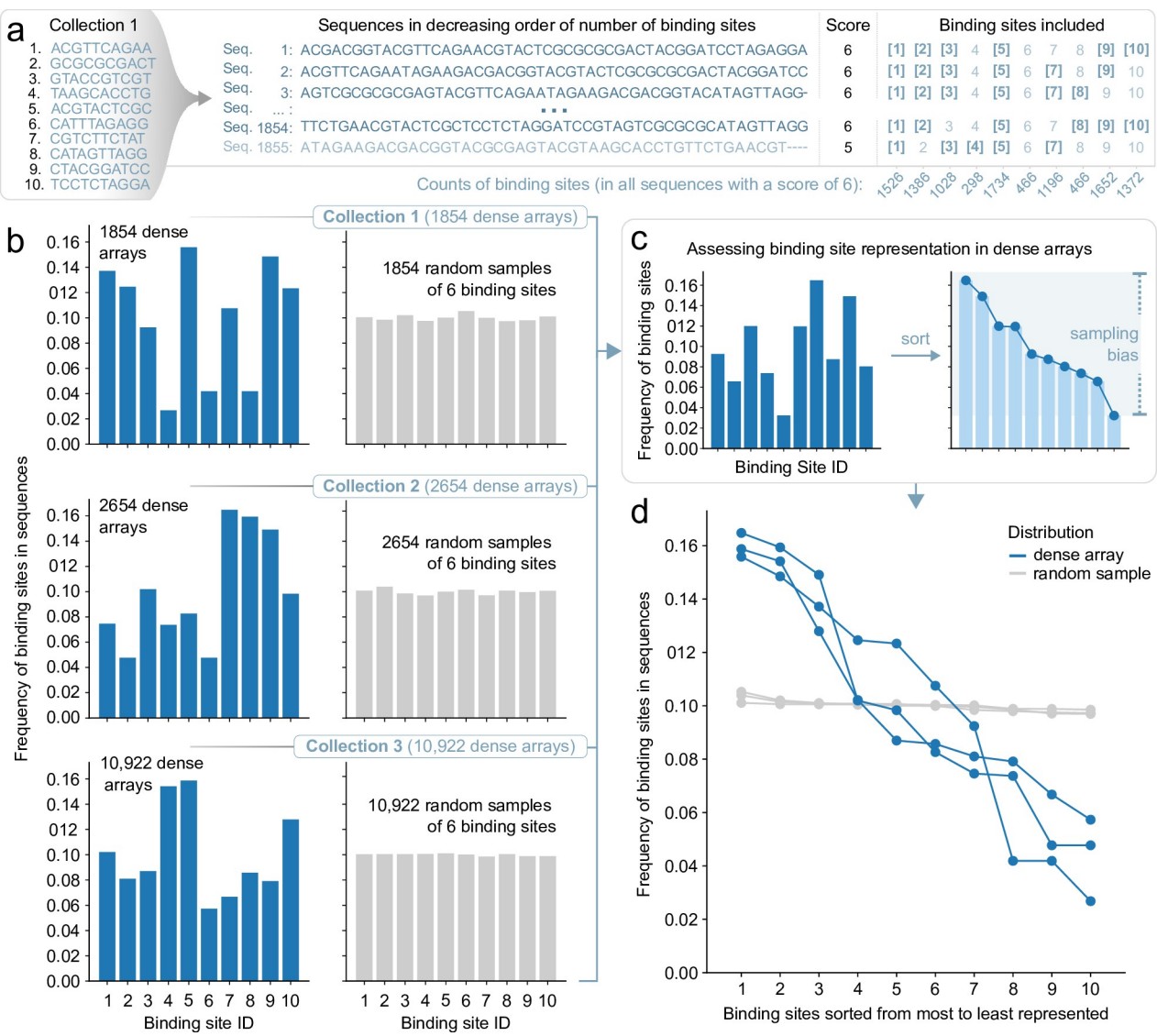

**Fig 3. Assessing binding site representation across dense arrays.** (A) For the |R| = 10 randomly generated binding sites of Collection 1 we found 1854 optimal sequences (each fitting 6 binding sites within L = 50 bp). We then tallied up the binding sites involved in these sequences. (B) The frequency of binding site distribution is represented as a histogram. Three random binding site collections of |R| = 10 binding sites, each of 10 base pairs in length, were used to create sequences of length L = 50 bp. The bar plots show the frequency at which each of the 10 binding sites was present in these sequences. Optimal solutions using the dense array approach with the SPP (blue) and using random samples of 6 binding sites (gray) are shown. (C) Sorting the histogram gives an idea of the heterogeneity of the distribution, where larger differences between the frequency of binding site usage correspond to larger sampling biases. (D) The histograms from (B) were first sorted by decreasing frequency, then assembled on this plot for comparison. The slope of the curve is an indicator of the diversity (or lack thereof) in the corresponding distributions.

## Generation of more representative solution sets

We address the issue of binding site representation bias by making a minor adjustment to our integer linear programming model. By maintaining a count of the binding sites included in previously generated solutions, we adjust the objective of the integer linear program to favor those binding sites that are underrepresented compared to the average. Concretely, we modify the objective to $\sum_{i\in[0,n],j\in[1,n]} c_{ij}\,x_{ij}$ where $c_{ij}$ is set to 1 for $j$ corresponding to binding sites more represented than the average so far, and to $1+\epsilon$ for binding sites less represented than the

average. The value of $\epsilon$ is irrelevant if it is sufficiently small to never allow the score of a solution with fewer binding sites to overcome the score of a solution with more binding sites (any strictly positive value that is smaller than $1/n$ meets this criterion). We apply this change after every generated solution. This change allows the solver to break ties between solutions in a different way each time. We refer to the original solution method as "solver order," because the specific solver implementation determines how ties are broken. We refer to our modified version of this approach that directs the algorithm towards the use of underrepresented sites as "diversity-driven order," because the addition of the weighting terms controls how the solver breaks ties when selecting which binding site to include (Fig 4A).

Whereas the solver order strategy leads to a pool of solutions that never represents every binding site equally (Fig 4B), we found that the diversity-driven strategy is effective at generating solutions in an order that transiently improves inclusivity in binding site representation (Fig 4C). In this example, $|R| = 10$ so equal representation of binding sites would have their frequency of appearance at 0.1. Both approaches ultimately result in the same binding site distribution because in the limit where they generate the maximum number of top-scoring solutions, the two strategies produce the same set of solutions. Critically, what is different is the order in which these solutions are returned before this maximum is reached. To summarize the results across the different solution methods, we calculated the entropy of the solver order and diversity-driven order strategies (Fig 4D). Higher entropy values correspond to more diverse solutions, where the theoretical maximum is $\log(|R|)$, reached only if all binding sites have been uniformly involved (with frequency $1/|R|$) in the solutions. The diversity-driven order solutions exhibited a rapid increase in entropy that was maintained across many rounds of binding site selection (Fig 4D), demonstrating its efficacy in generating solutions with broad representation. The diversity-driven order also produced higher entropy distributions than the solver order for collections with size differences among binding sites (S5 Fig).

## Incorporating positional constraints for promoter design

We next incorporated more constraints into the solver to facilitate the inclusion of sequence elements at fixed positions. As an example, we focused on the design of bacterial promoters. Sigma factor recognition sites are crucial elements in bacterial promoters, and they typically appear around -35 and -10 base pairs away from the transcriptional start site [56]. Incorporating constraints at specific locations forces the solver to place user-defined sites within dense arrangements of binding sites on the forward strand (Fig 5A and 5B, see Methods for the implementation of the constraints). We designed the constraints to be flexible enough that the user can specify arbitrary ranges, for example specifying that the downstream element should start between the -9 and -11 positions relative to the end of the sequence. This option for incorporating positional flexibility helps the solver to find feasible solutions that align with user-defined constraints and allows for the specification of multiple fixed sequence elements (or pairs thereof) to be positioned at defined proximal distances from one another. For example, using this strategy, the solver can design dense arrays that contain tightly interlocked primary and alternative sigma factor recognition elements (Fig 5C and 5D). In nature, promoters often overlap, resulting in transcription start sites that are shared or close to each other, each linked to specific sigma factor recognition sites upstream [1]. The positioning and spacing of these sites is shaped by the biophysical constraints of the corresponding sigma factor [57]. Incorporating multiple sigma factor pairings into synthetic promoters helps sustain transcriptional strength across diverse environments and transient stress responses, which can be valuable for biotechnological applications [26,58]. As an example, we incorporated three distinct promoters within a single dense array, featuring consensus sequences for $\sigma^D$, $\sigma^S$, and $\sigma^N$. These sigma

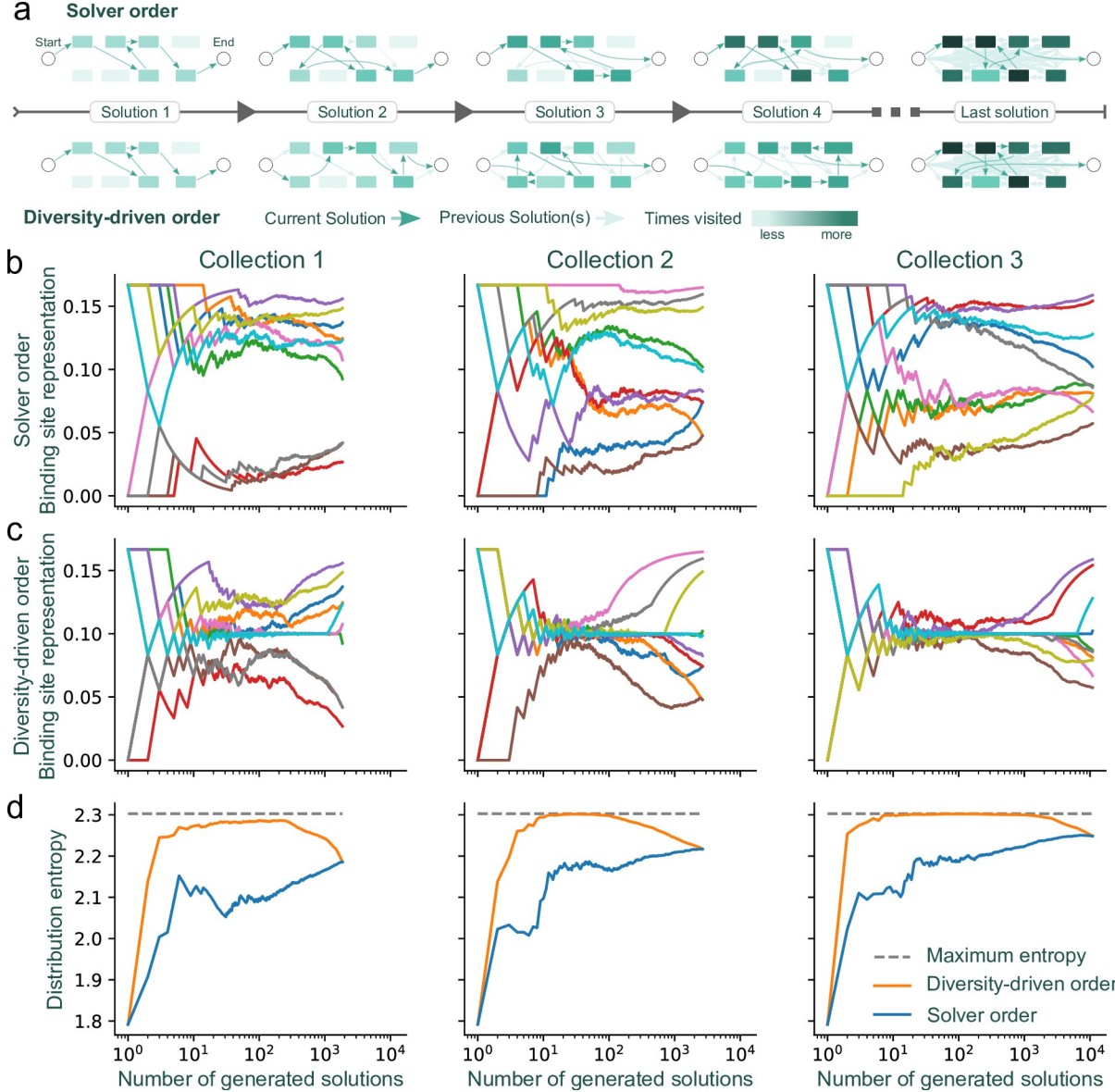

**Fig 4. A minor modification to the model fosters a diverse representation of binding sites as solutions are returned.** (A) Schematic representing binding site selection frequencies using solver order (top) or diversity-driven order (bottom). The color intensity of each vertex represents the absolute counts with which it was included in a solution. (B) Solver order, using the Gurobi solver, and (C) diversity-driven order approaches generate different frequencies of binding site representation. Each color represents a different binding site ($|R| = 10$ binding sites). The three columns represent the three binding site collections presented in Fig 3, with respective number of top-scoring solutions of 1854, 2654, and 10,922. (D) The distribution entropy of the solver order and diversity-driven order strategies. In all cases, the entropy of the diversity-driven order is the same or higher than the entropy of the solver order.

factors are linked with housekeeping genes, stationary phase and the general stress response, and nitrogen metabolism, respectively. This combination enables the downstream genes of this synthetic promoter to theoretically respond to variations in any of these conditions. With fixed sequence lengths and positional constraints, we used random sequences to represent binding sites and generated dense arrays. The solver successfully returned solutions that placed specified binding sites in predetermined positions (Fig 5D).

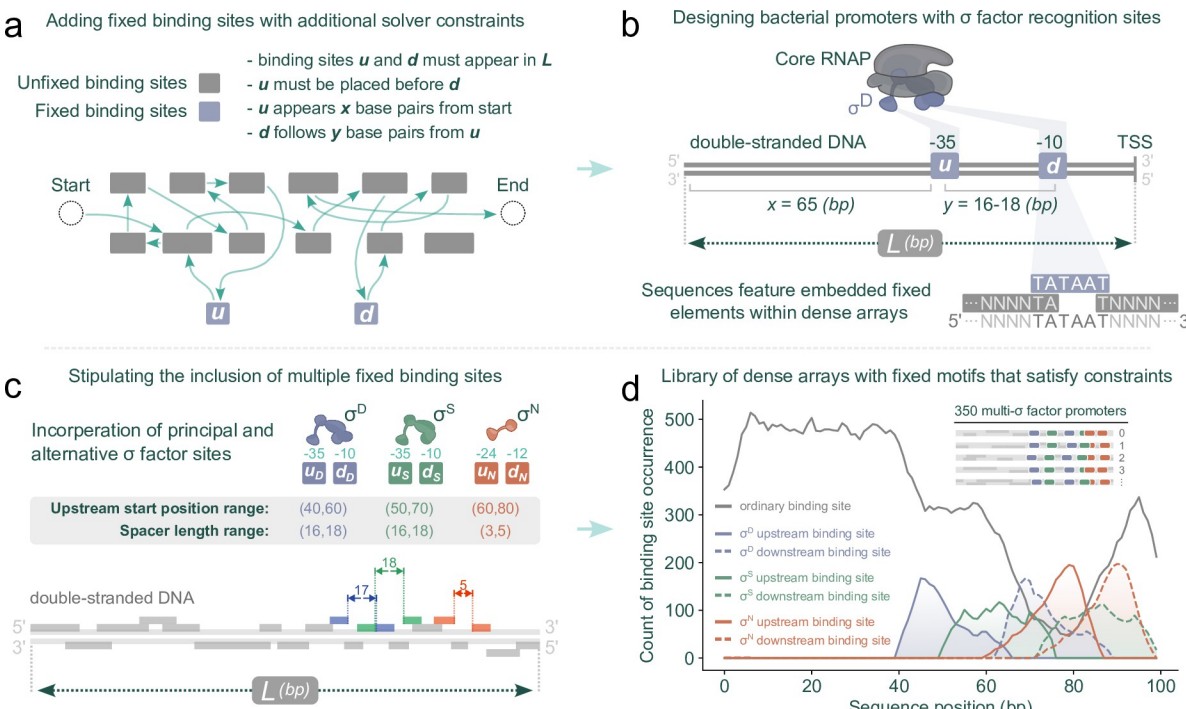

**Fig 5. Stipulating positional constraints to design bacterial promoters.** (A) By applying additional constraints, the solver is guided to create sequences with user-defined binding sites at predetermined locations. u, upstream site; d, downstream site. (B) This technique enables interweaving of dense binding site regions around -35 and -10 sigma factor recognition sites. (C) Extending this, the solver can incorporate both primary and alternative sigma factor sites, placed within densely packed binding sites. (D) Summary of results from 350 binding site collections. Each collection contained random sequences to represent binding sites and was used to generate an output sequence of $L = 100$ base pairs. Each collection contained $|R| = 20$ binding sites, with sizes uniformly chosen between 5 to 15 base pairs. Alongside the ordinary binding sites in each collection, six additional fixed binding sites were introduced with varying constraints: TTGACA/TATAAT, with its upstream position within 40–60 base pairs from the beginning of the sequence and a spacer length of 16–18 base pairs; TTGACA/TATACT, with its upstream position within 50–70 base pairs and also with a spacer length of 16–18 base pairs; and TGGCAGG/TTGCA, with its upstream position within 60–80 base pairs, but with a shorter spacer length of 3–5 base pairs. The total number of random binding site collections generated was 700: in half of the cases, the solver reported that no solution satisfying the constraints existed.

Transcription factor binding sites play varied roles in gene expression; for instance, some facilitate activation while others cause repression, depending on the activity of their associated regulator or their relative position to a transcription start site. Notably, it has been thought that binding sites for activators typically lie upstream of the -35 sigma factor recognition element, with their effectiveness diminishing if they are placed further downstream [23,56]. To demonstrate how the model can accommodate this, we classified random sequences that represented binding sites as being associated with either activators or repressors, and then adjusted the model to position activator sites more upstream and repressor sites more downstream (Fig 6A, see Methods for the implementation). Modifying the linear objective in this way provides an additional layer of optimization to the solver, where otherwise equally scored solutions (e.g., two dense arrays that each can fit 19 binding sites) will be differentiated based on where activator sites and repressor sites appear in their sequences (Fig 6B).

## Discussion

Dense arrangements of DNA-protein binding sites within nucleotide sequences can collectively influence downstream transcription rates or initiate biomolecular interactions. Despite the prevalence of such sequences in nature, rapidly generating large stretches of DNA with

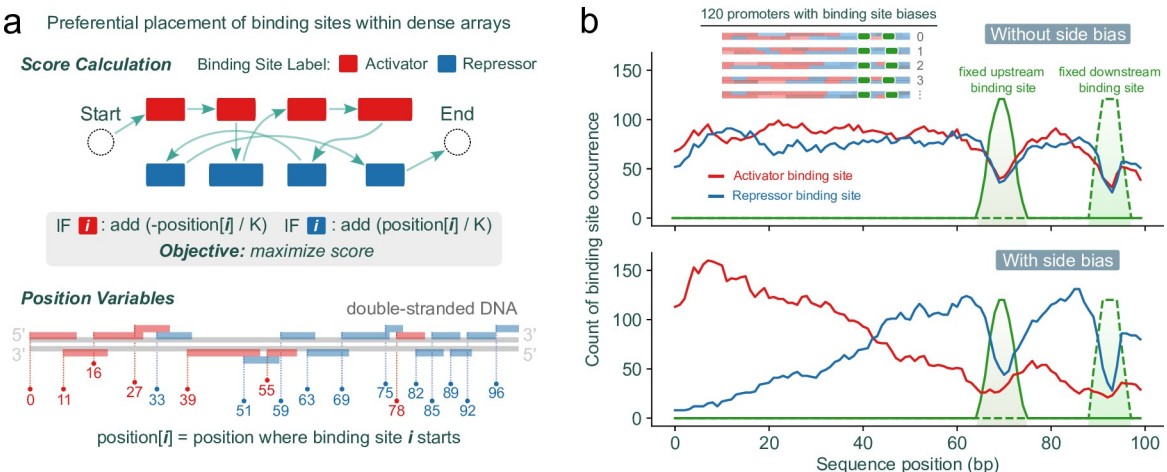

**Fig 6. Modifying the linear integer programming model objective to include a position bias for activator sites and repressor sites.**
(A) Classifying binding sites as either activator or repressor sites and then modifying the model to favor said sites by minimizing/
maximizing the position where they appear in the sequence. (B) The histogram was generated from data obtained from 120 random
binding site collections, each consisting of $|R| = 20$ binding sites. These binding sites, classified randomly either as activator sites or
repressor sites, varied in length from 5 to 15 base pairs, and were used to assemble sequences of $L = 100$ base pairs. Additionally, two
6-base pair sites, representing sigma factor recognition elements, were included with specific constraints: the downstream element was
positioned between positions 89 and 91 (corresponding to the -10 site), with a spacer of 16–18 base pairs (following the -35 site).

overlapping binding sites remains a challenge. Here, we present a computational method to
pack nucleotide sequence binding sites into dense arrays (Fig 1). We initially formulated the
task as a problem that we denote as the nucleotide SPP, which we showed to be NP-hard. The
SPP takes its place within the classical NP-hard problems, in that its relation with the Shortest
Common Superstring mirrors the relation of other pairs of classic NP-hard problems, such as
Maximum Coverage and Set Cover, Multiple Knapsack and Bin Packing, or Orienteering
Problem and Travelling Salesman Problem. We then reduced the task to an Orienteering
Problem with integer distances, which we expressed as a problem that can be solved using inte-
ger linear programming—a highly expressive framework that is amenable to the inclusion of
user-defined constraints, while also enabling the use of modern and efficient solvers. Although
the task is still NP-hard, this approach allowed us to dramatically increase the computational
efficiency of obtaining solutions over brute force approaches. Importantly, for biologically
realistic parameters (e.g., sequence lengths of $L = 50$–300 base pairs with several dozen binding
sites), this formulation can be solved in seconds (Fig 2). In contrast, a greedy polynomial-time
method was only able to reach the optimal solution when the problem was at its easiest (e.g.,
with few binding sites in a long sequence involving little to no compression (S1 Fig)).

By enumerating the best solutions in order, we found that there can be a multitude of
sequences that achieve optimal packing for a given binding site collection and sequence size
(Fig 3). However, there is a strong bias in the representation of binding sites in the set of opti-
mal solutions, where shorter binding sites appear more frequently (S2 Fig) and collection-spe-
cific effects impact binding site representation (Fig 3). While we highlight here how sampling
binding sites from a collection can produce a library of distinct sequence variants, the problem
of sampling is independent of the SPP. Sampling efforts can be customized in several ways to
generate sets of sequences that meet specific binding site representation requirements. This
can be achieved by iteratively modifying the integer linear programming model itself, such as
by assigning different weights to various binding sites in the solution score, as demonstrated in
our approach (Figs 4 and S4). We also demonstrate how the solver can incorporate specific

sequence elements at fixed positions within the dense arrays through additional constraints. This method allows for the strategic placement of binding sites (or pairings thereof) at defined locations, enabling the scalable design of bacterial promoters with intricate binding site architectures (Fig 5). Moreover, solutions can be further refined by adding solver incentives based on the functional role of binding sites, such as being associated with either activators or repressors. This approach can guide preferential placement of binding sites either upstream or downstream within the dense array (Fig 6). Notably, we have observed that solve time increases as more constraints and incentives are applied. Tasks like generating dense arrays constrained only by binding site collection size $|R|$ and sequence length $L$ (as shown in Fig 2) typically resolve in seconds. In contrast, more complex arrays stipulated with "diversity-driven order" (Fig 4) or "positional bias" (Fig 6) exhibit greater variability in solve time, ranging from seconds to tens of seconds. Occasionally, the solver may time out—for example due to the imposition of too many constraints, resulting in no viable solutions for the given binding site collection or sequence length.

It is also worth noting that the presented method can extend beyond cis-regulatory regions and nucleotides. It can, in theory, densely pack strings made from any set of characters. Therefore, it is amenable to forming continuous stretches of virtually any primary biological sequence. The advantage of this graph-based approach is expected to increase as the alphabet size increases because our formulation is independent of the size of the alphabet, in contrast to methods modeling each letter individually. Beyond biological applications, the SPP introduced here could find use cases in data compression, as seen with its variant the SCS problem [59,60].

Further work could explore designing constraints that account for additional structural and proximal factors influencing gene expression. This includes exploring how a regulator's bound, helical positioning on the DNA influences its distance to RNA polymerase, thus affecting its regulatory effectiveness [56]. Another consideration for bacterial promoter design is the prevalence of transcription factors that operate as homodimers, which form binding sites that resemble "spaced dyads," characterized by less conserved middle regions [56]. The dense arrangement of these dyads may present additional constraints and challenges to the solver, necessitating more nuanced and sophisticated design strategies. However, many binding sites are defined as fuzzy, in contrast to the strict binding sites associated with restriction enzymes, thus many arrangements of densely packed binding sites would likely give rise to nuanced and complex dynamics in cis-regulatory regions. In addition, existing tools in the field of computational promoter design, such as predictive DNA sequence-to-activity models, could be applied to dense arrays generated by the SPP—ascribing features such as promoter likeness or expression strength, with potential applications for *in silico* directed evolution.

The design and application of DNA sequences composed of discrete binding sites is reliant on the availability of accurate and biologically relevant binding site annotations. Fortunately, continued characterization of DNA-protein binding profiles across organisms, sourced from techniques like ChIP-seq, DAP-seq, ATAC-seq, and HT-SELEX, will support the curation of binding sites associated with proteins and cellular mechanisms. It is worth noting that there is no definitive threshold for classifying a sequence as a transcription factor binding site. These sites exist on a continuum, with affinities ranging from so low as to be negligible, to so high that the transcription factor is nearly always bound [61]. Indeed, protein-DNA binding data for transcription factors typically encompass a range of binding peaks rather than a single sequence. These sequences can be affiliated with varying degrees of binding affinity, determined by factors such as experimental enrichment [62] or their similarity to a consensus sequence. In practice, when multiple binding sites with labeled affinities are available for a transcription factor, one can choose among these binding sites as inputs for the SPP method.

This flexibility allows for the tailoring of output solutions that meet specific design criteria or assumptions. However, comprehensive binding affinity data is available for only a limited set of transcription factors, and representative consensus sequences do not necessarily equate to the highest thermodynamic binding affinity [63,64]. Despite the uncertainties, the SPP is a practical tool for studying these questions through forward engineering. Equally applicable to large-scale libraries and more focused studies, the nucleotide string packing approach we introduce here serves as a resource for designing information-dense promoter sequences, which could then be used to map how intricate binding site configurations influence gene expression or biomolecular mechanisms.

## Methods

### Formulation of the SPP as an integer linear program

Given that the SPP is NP-hard, there is no known algorithm that can solve it in a time that is polynomial in the size of the input, making solving medium to large instances impractical with naïve methods. However, many NP-hard problems have practical significance, which has led to the development of optimized solvers [46], designed to manage the combinatorial explosion and return optimal solutions (as time and memory constraints permit). Although these solvers might not return solutions in large instances within their allocated time and memory limits, they can offer surprising efficiency for smaller problems. Many numerical solvers have been specifically developed to tackle the NP-hard integer linear programming problem, which restricts linear programming to integer variables. We formulated the SPP as an integer linear programming problem by reducing it to a variant of the Travelling Salesman Problem (TSP) known as the Orienteering Problem (OP) [44,47,48]. The TSP, a classic example of an NP-hard problem, involves determining the shortest possible route that visits every vertex of a graph. In the OP, there is an allocated time budget to traverse the graph, and the goal is to visit as many distinct vertices as possible within that time budget. The OP's relation to the TSP mirrors the SPP's relation to the SCS problem. In fact, this relationship is shared by several other pairs of classical NP-hard problems, where one problem consists of minimizing the cost of covering all possible elements (e.g. Set Cover, Bin Packing, Traveling Salesman, Shortest Common Superstring), and the other consists of maximizing the number of elements covered within an arbitrary given cost (respectively: Maximum Coverage, Multiple Knapsack, Orienteering Problem, and the String Packing Problem which to our knowledge had not been named yet). Note that although similar, these pairs of problems are not equivalent, nor can they be solved or approximated with the same efficiency. For instance, while Maximum Coverage admits a constant-factor polynomial-time approximation algorithm [65], it has been demonstrated that Set Cover cannot be approximated to $(1 - o(1)) \ln n$ unless P = NP [66]. We can map any SPP to an OP by representing binding sites as vertices on a complete directed graph.

We define a metric $d_{ij}$ to represent the cost associated with going from one nucleotide string to another in the graph. Specifically, the edge that connects string $a$ and string $b$ is given a natural integer weight, which represents the offset needed for the prefix of string $b$ to match the suffix of string $a$ (Fig 1B). In other words, it describes the minimal number of spaces that string $b$ needs to be shifted to the right to coincide with and cover the end of string $a$. For example, the directed edge from AGC to CAG is weighted 2, while the reverse edge is weighted 1. If no suffix of $a$ matches a prefix of $b$, then we consider that the empty suffix of $a$ matches the empty prefix of $b$, such that CGT to CAG is weighted 3. Importantly, a suffix of $a$ needs to match a prefix of $b$, such that $b$ overhangs at the end of $a$ (possibly by 0 characters), but cannot stop before the end of $a$, for example, the directed edge from AGCAG to GC must be weighted 4,

not 1 (1 shift does not allow GC to overhang in the end of AGCAG). This requirement allows the path to resume from the last vertex visited (here, GC) without any additional constraints. It also makes the metric $d_{ij}$ satisfy the triangle inequality. Finally, we create two additional vertices: a start vertex with distance 0 to all other vertices, and an end vertex, such that the distance of any binding site to the end vertex is the length of this binding site (Fig 1C). Maximizing the number of binding sites included within a string of length $L$ is then equivalent to maximizing the number of graph vertices visited between the start and end vertices, with a sum of edge weights not larger than the total budgeted length $L$. This corresponds to solving the OP on this graph. Because all the distances are integers, this problem can be modeled as an integer linear program. We adapt the Miller–Tucker–Zemlin TSP formulation [67] to consider the specificities of our problem, as follows:

## Variables

We represent a path in the graph with binary variables. For every edge between two binding sites, we create one binary variable to indicate if this edge is taken, that is, if the second binding site comes directly after the first one in the final sequence (with the specified overlap). This creates $n^2+n+1$ binary variables: with the index 0 representing both the start and end vertex, we have $n(n-1)$ edges between internal vertices ($x_{ij}$ with $i \neq j$), $n$ edges between the start and all internal vertices ($x_{0j}$), $n$ edges between all internal vertices and the end vertex ($x_{i0}$), as well as one edge between the start and the end ($x_{00}$) (taken if and only if no binding sites can fit in the solution, that is, if the sequence length $L$ is shorter than all binding sites).

## Objective

The solver's task is to maximize the number of vertices visited by the path in the graph. This is equivalent to maximizing the sum of binary variables representing edges that enter an internal vertex (not the end vertex): $\sum_{i \in [0,n], j \in [1,n]} x_{ij}$. The following constraints impose that $x_{ij} = 1$ if and only if the path contains the edge going from $i$ to $j$.

## Constraints

1. The path must start at the start vertex and end at the end vertex: $\sum_{i \in [0,n]} x_{i0} = 1$ and $\sum_{j \in [0,n]} x_{0j} = 1$.

2. Vertices cannot be visited more than once: $\forall i \in [1,n], \sum_{j \in [0,n]} x_{ij} \leq 1$ and $\forall j \in [1,n], \sum_{i \in [0,n]} x_{ij} \leq 1$.

3. Any given internal vertex is entered and exited the same number of times: $\forall k \in [1,n], \sum_{i \in [0,n]} x_{ik} = \sum_{j \in [0,n]} x_{kj}$.

4. The total length of the path is not longer than the desired final sequence: $\sum_{i \in [1,n], j \in [0,n]} d_{ij} \, x_{ij} \leq L$ where $d_{ij}$ represents the metric described above.

5. Miller–Tucker–Zemlin subtour elimination constraints introduce one additional integer variable per binding site, $\forall i \in [1,n], \ u_i \in [1,n]$, and $n(n-1)$ additional constraints: $\forall i \in [1,n], \ \forall j \in [1,n], \ u_i - u_j + 1 \leq n(1 - x_{ij})$. These additional variables and constraints implement a counter that must increase along the path, which prevents subtours (loops disconnected from the main path).

Reconstructing the solution from the $x_{ij}$ variables yields solutions such as those pictured in Fig 1D. The expressivity of integer linear programming allows us to modify this base model to

include additional constraints and specifications. For example, double-stranded optimization (allowing binding sites to appear not only in one strand but also as their reverse-complement on the other) can be implemented by duplicating every vertex, reverse-complementing the sequences, and modifying constraint 2 to prevent the algorithm from visiting both a binding site and its reverse complement. Note that while the single-strand version of the SPP mirrors the OP, the double-strand version of the problem is not strictly an OP, as it involves additional constraints such as restricting the inclusion of both a binding site and its reverse complement. However, both single and double stranded versions can be solved efficiently using integer linear programming. We implemented the SPP integer linear programming model with the OR-Tools optimization toolbox [46], which is able to call open-source (CBC [68] or SCIP [69]) or commercial (Gurobi [70]) integer linear programming solvers. Unless otherwise noted, we use the Gurobi solver due to its superior speed (S1 Table).

### Implementation of the promoter constraints

Promoter constraints are formulated in terms of absolute positions in the DNA sequence: these positions must then be encoded in the linear integer model. We achieve this with a set of $n$ integer variables with values from 0 to L-1 (both included), and the following linear constraints:

$$\forall i \in [0, n], \ \forall j \in [1, n], \ i \neq j, \ d_{ij} x_{ij} - (L-1)(1 - x_{ij}) \leq position_j - position_i \leq d_{ij} x_{ij}$$
$$+ (L-1)(1 - x_{ij})$$

where $position_0 = 0$ (the position of the empty start vertex).

This follows a similar approach as the subtour elimination constraints, forcing $position_j - position_i$ to be equal to $d_{ij}$ if and only if $x_{ij} = 1$ (otherwise, any value is possible).

With these position variables defined, we implement the promoter position constraints as follows:

- If binding site $k$ corresponds to a fixed binding site, it must appear in the sequence: $\sum_{i \in [0,n], \ i \neq k} x_{ik} \geq 1$

- If binding site $k$ corresponds to a fixed binding site that needs to start between positions $\alpha$ and $\beta$, then $\alpha \leq position_k \leq \beta$

- If binding sites $k$ and $l$ are the upstream and downstream elements of a promoter where the spacer length is constrained to be between $\alpha$ and $\beta$ base pairs: $\alpha \leq position_l - length_l - position_k \leq \beta$ with $length_l$ being the length of binding site $l$

### Implementation of the side bias

The side biases (introducing binding sites with upstream or downstream preferences) can be implemented using the position variables described in the previous subsection. We add the term $\pm position_i / K$ to the score function if binding site $i$ has an upstream preference ($-$) or a downstream preference ($+$). These terms are added to an integer-step scoring function (the number of binding sites in the sequence). The value of the strictly positive constant $K$ is thus irrelevant as long as the sum of all these terms cannot take a value outside of $[-0.5, 0.5]$: this ensures that two solutions with different numbers of binding sites cannot be reordered.

### Supporting information

**S1 Fig. Performance of the greedy approximation algorithm.** We plotted the number of binding sites that the approximation algorithm managed to fit into a sequence of a given

length, $L$, normalized by the optimal solution. We repeated each scenario 10 times with random binding sites of uniform random lengths between 5 and 15 base pairs. Shaded regions represent the bootstrapped 95% confidence interval around the mean.
(TIF)

**S2 Fig. Frequency of binding site usage as a function of binding site length.** Ten binding site collections were randomly generated with 10 binding sites each, where there is one binding site of each length from 5 to 14 base pairs. All of the top-scoring solutions for a sequence length $L = 50$ were generated every time: for one of the 10 binding site collections, 4 binding sites were able to be fit at best, with 7184 ways to do so. For the nine others, 5 binding sites were able to be fit at best, with 26, 72, 150, 184, 206, 278, 418, 664, 762 ways to do so. The shaded region represents the bootstrapped 95% confidence interval around the mean.
(TIF)

**S3 Fig. Within binding sites of the same size, the ones with the highest average overlap (lowest average distance $d_{ij}$) with the others are disproportionately represented in the top-scoring solutions.** The blue line is the average linear regression line, the shaded region is the 94% credible interval of the mean.
(TIF)

**S4 Fig.** Solver order with (A) Gurobi, (B) SCIP, and (C) CBC solvers. (D) Diversity-driven order control strategies. Details on the solve times for the different solvers are available in S1 Table. (E) The diversity-driven order approach generally produces higher entropy distributions despite the bias in binding site representation. Note that data from Fig 4 for the Gurobi solver, diversity-driven order, and entropy plots are replicated here for ease of comparison.
(TIF)

**S5 Fig.** The solver order (A) and diversity-driven order (B) strategies applied to heavily biased libraries, where binding sites have different sizes. Every library is made of 10 randomly generated binding sites, one of size 5 base pairs, one of size 6, etc., until size 14. As the full distribution attests (rightmost point of the graphs), some binding sites are present in almost all top-scoring solutions, while some others are present in none. Despite this, the diversity-driven order approach generally produces higher entropy distributions (C). The solver used here was Gurobi.
(TIF)

**S1 Table. Time to enumerate all top-scoring solutions using solver order, for three different solvers.** For solve times $> 20$ h, we stopped computation at this point.
(PDF)

## Acknowledgments

We would like to thank Hang Zhou and Rayan Chikhi for helpful comments on the manuscript.

## Author Contributions

**Conceptualization:** Eric J. South.

**Formal analysis:** Virgile Andreani.

**Funding acquisition:** Mary J. Dunlop.

**Investigation:** Virgile Andreani.

**Methodology:** Virgile Andreani.

**Project administration:** Mary J. Dunlop.

**Software:** Virgile Andreani, Eric J. South.

**Supervision:** Mary J. Dunlop.

**Validation:** Eric J. South.

**Visualization:** Eric J. South.

**Writing – original draft:** Virgile Andreani, Eric J. South, Mary J. Dunlop.

**Writing – review & editing:** Virgile Andreani, Eric J. South, Mary J. Dunlop.

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
