## [Decision Letter · Decision Letter 0]

10 Apr 2024

Dear Dr Dunlop,

Thank you very much for submitting your manuscript "Generating information-dense promoter sequences with optimal string packing" for consideration at PLOS Computational Biology.

As with all papers reviewed by the journal, your manuscript was reviewed by members of the editorial board and by several independent reviewers. In light of the reviews (below this email), we would like to invite the resubmission of a significantly-revised version that takes into account the reviewers' comments.

The reviewers appreciate the goal to computationally generate promoters with densely packed binding sites. They raise several points to discuss, which will enhance the paper. Furthermore, both reviewers suggest that experimental validation would significantly strengthen the manuscript, so please consider whether this is possible.

We cannot make any decision about publication until we have seen the revised manuscript and your response to the reviewers' comments. Your revised manuscript is also likely to be sent to reviewers for further evaluation.

Sincerely,

Stefan Klumpp

Academic Editor

PLOS Computational Biology

Stacey Finley

Section Editor

PLOS Computational Biology

Both reviewers suggest that experimental validation would significantly strengthen the manuscript, so please consider whether this is possible.

Reviewer's Responses to Questions

**Comments to the Authors:**

Reviewer #1: Summary of manuscript:

In the manuscript entitled “Generating information-dense promoter sequences with optimal string packing,” the authors described solution methods to create promoter sequences that contain many transcription factor binding sites in a specified (typically short) length of DNA. The final solver developed for this task is available online, and the ability to generate promoters with densely packed binding sites could be of general interest to the synthetic biology or cell engineering communities. However, the functionality of at least one promoter library must be shown to demonstrate the expected value of this novel solution method. Demonstration of bacterial promoter library function would suffice.

Other comment:

(1) It is often unclear what the library size is for each generated library, referring to the number of sequences that would need to be screened to test for function (number of sequences generated), instead of the author’s definition of library size [R] = number of binding sites. This information is necessary to gauge the usefulness of each generated library, as screening 10^6 promoters for function might be possible in one system, while testing 10^1 might be more feasible in another. As the manuscript sells the SSP method for promoter library generation, discussion of the feasibility of testing the generated libraries is warranted.

(2) Promoter library sequences, the full sequences in addition to the inputted binding sites, should be included in the supplement or extended data, primarily for libraries that are expected to have function like the bacterial promoter libraries.

Reviewer #2: Natural promoter regions may contain many overlapping binding sites of transcriptional factors, affecting transcription initiation rates. Despite the common occurrence of overlapping binding sites in nature, the rapid artificial design of nucleotide sequences with many overlapping sites remains a challenge. In this paper, the authors propose a computational approach for designing nucleotide sequences with densely packed DNA-protein binding sites, termed the Nucleotide String Packing Problem (SPP). They first demonstrate that the SPP problem is NP-hard, and thus simplify the problem into Orienteering Problem with integer distances, which can then be efficiently solved using various open-source and commercial solvers. The authors subsequently explore many possibilities of the method in the design of bacterial promoters.

Suggestions provided are as follows:

1.Regarding the issue of bias in solutions, the authors attempt to explore the effects of binding site size and sequence on bias, while briefly mentioning the potential impact of different solvers due to their different internal algorithms. However, the explanation for the effects of binding site sequences and different solvers is not sufficiently clear. For the effect of sequences, one approach could be to investigate the influence of bias from the perspective of sequence overlap. Additionally, exploring different solvers and observing their specific effects on bias, if any, could also be attempted here.

2.The article mentions that "Meanwhile, generative AI techniques are starting to show promise in emulating the complexity of context-dependent promoters (31–37). However, these models often struggle with interpretability, and fine-tuning them to include or exclude specific binding sites still requires specialized expertise (38)." However, in practice, the method designed in this article may rely more heavily on specialized expertise, as understanding different binding sites may involve complex processes. Additionally, whether existing expert knowledge is sufficient to generate binding site libraries consistent with natural promoters is worth discussing. It is recommended that the authors provide a clearer explanation in this regard.

3.Furthermore, due to the ambiguity in determining binding sites in biology and the variation in protein-motif binding across different biological states, whether more densely distributed binding sites correspond to a more suitable promoter is still worth considering. It is hoped that more discussion on this aspect will be provided in the Discussion section.

4.If experimental conditions permit, synthesizing designed promoter sequences and subsequently measuring the strength of artificially designed promoters using methods such as fluorescence protein assays would enhance the persuasiveness of the article.

5. The introduction part lacks a comprehensive overview of the categories of computational methods related to promoter design, and there are additional types of computational methods relevant to promoter design that should be introduced. For example, some promoter strength predictive models (classification/regression), which may play a crucial role in in silico directed evolution.

6.Minor issue: The title of Figure 4 is not bold, inconsistent with other figures.

**Have the authors made all data and (if applicable) computational code underlying the findings in their manuscript fully available?**

Reviewer #1: **No: **I don't think the data generated is available

Reviewer #2: Yes

PLOS authors have the option to publish the peer review history of their article (what does this mean?). If published, this will include your full peer review and any attached files.

Reviewer #1: No

Reviewer #2: No
---

## [Decision Letter · Decision Letter 1]

25 Jun 2024

Dear Dr Dunlop,

We are pleased to inform you that your manuscript 'Generating information-dense promoter sequences with optimal string packing' has been provisionally accepted for publication in PLOS Computational Biology.

Best regards,

Stefan Klumpp

Academic Editor

PLOS Computational Biology

Stacey Finley

Section Editor

PLOS Computational Biology

Reviewer's Responses to Questions

**Comments to the Authors:**

Reviewer #2: The authors have addressed my concerns, therefore I think this article can be accepted.

**Have the authors made all data and (if applicable) computational code underlying the findings in their manuscript fully available?**

Reviewer #2: None

PLOS authors have the option to publish the peer review history of their article (what does this mean?). If published, this will include your full peer review and any attached files.

Reviewer #2: No

---

## [Editor Report · Acceptance letter]

11 Jul 2024

PCOMPBIOL-D-24-00208R1 

Generating information-dense promoter sequences with optimal string packing

Dear Dr Dunlop,

I am pleased to inform you that your manuscript has been formally accepted for publication in PLOS Computational Biology. Your manuscript is now with our production department and you will be notified of the publication date in due course.

With kind regards,

Zsofia Freund
